# Cerebral cortical thinning in Parkinson's disease depends on the age of onset

**Kazuhide Seo**[1]*, **Ichiro Matunari**[2], **Toshimasa Yamamoto**[1]

**1** Department of Neurology, Saitama Medical University, Saitama, Japan, **2** Department of Radiology, Division of Nuclear Medicine, Saitama Medical University, Saitama, Japan

* k_seo@saitama-med.ac.jp

## Abstract

Patients with older-onset Parkinson's disease (PD) have more severe motor symptoms, faster progression, and a worse prognosis. The thinning of the cerebral cortex is one of the causes of these issues. Patients with older-onset PD manifest more extended neurodegeneration associated with α-synuclein deposition in the cerebral cortex; however, the cortical regions that undergo thinning are unclear. We aimed to identify cortical regions with different thinning depending on the age of onset in patients with PD. Sixty-two patients with PD were included in this study. Patients with PD onset at <63 years old were included in the early or middle-onset PD group, and those with PD onset at >63 years old were included in the late-onset PD (LOPD) group. Brain magnetic resonance imaging data of these patients were processed using FreeSurfer to measure their cortical thickness. The LOPD group displayed less cortical thickness in the superior frontal gyrus, middle frontal gyrus, precentral gyrus, postcentral gyrus, superior temporal gyrus, temporal pole, paracentral lobule, superior parietal lobule, precuneus, and occipital lobe than the early or middle-onset PD group. Compared with patients with early and middle-onset PD, elderly patients displayed extended cortical thinning with disease progression. Differences in the clinical manifestations of PD according to the age of onset were partly due to variations in the morphological changes in the brain.

## Introduction

Parkinson's disease (PD) is a neurodegenerative disorder caused by the loss of dopaminergic neurons in the substantia nigra; however, its clinical phenotype is complex and heterogeneous [1,2], with differing motor and non-motor symptoms, age of onset, and rate of progression in individual patients. Among these factors, the age of onset may determine the clinical phenotype [3–6]. The older the patient, the more severe the motor symptoms are, particularly bradykinesia, gait disturbance, postural instability [7,8], and disease progression. Furthermore, older patients are more likely to develop dementia, hallucinations, insomnia, constipation, and olfactory disturbances [8,9] and have a poorer prognosis [10]. These findings suggest a strong relationship between aging and cerebral cortex degeneration. The specific degeneration of dopaminergic neurons in the substantia nigra is a pathological hallmark of PD; moreover, neurodegeneration occurs in cortical areas with disease progression. The appearance of Lewy body pathology in the cerebral cortex begins in the anterior medial temporal cortex, followed

**Data Availability Statement:** All relevant data are within the paper and its Supporting Information files.

**Funding:** The authors received no specific funding for this work.

**Competing interests:** The authors have declared that no competing interests exist.

by the higher sensory association areas, prefrontal cortex, primary association sensory areas, premotor areas, and primary sensory and motor areas [11]. Additionally, neurodegeneration due to cortical α-synuclein deposition is more common in elderly patients with PD [12]. The cortical regions that are more severely affected in such patients have not yet been identified. This necessitates clarifying the relationship between the clinical characteristics and pathophysiology in elderly patients with PD.

FreeSurfer is an open-source software that is used for quantifying and analyzing brain magnetic resonance imaging (MRI) images. A quantitative analysis of brain MRI images was previously performed by manually setting the region of interest for each slice. This requires considerable time and effort, but it cannot analyze the whole brain. With the development of computer technology, researchers have created methods and software for quantitatively analyzing brain MRI. Voxel-based morphometry (VBM) [13] was established in 2000 and is a semi-automatic method that can analyze the whole brain in voxel units. It has been used in brain MRI studies instead of manual methods. FreeSurfer is equipped with surface-based morphometry (SBM), which is specialized for the cortical analysis of brain MRI quantitative analysis methods [14]. SBM is used in brain MRI research and can measure the cortical thickness, cortical surface area, and gyrus in the whole brain unlike VBM.

The purpose of this study was to identify cortical regions with reduced thickness in patients with PD depending on the age of onset by using FreeSurfer as a tool for SBM.

## Materials and methods

### Subjects

We enrolled 62 patients who met the clinical diagnosis criteria of PD by the Movement Disorder Society [6] and who visited Saitama Medical University Hospital between January 2000 and September 2019. This study was a retrospective study and was approved by the ethical review committee of Saitama Medical University (Approval No. 19074.01). Patients with a history of heavy alcohol consumption; anticancer drug use; psychiatric disorders; neurodegenerative diseases not including PD; and intracranial lesions, such as cerebral infarction, cerebral hemorrhage, and head trauma with the loss of consciousness, were excluded from the study. Patients had one or more MRI examinations under the imaging conditions described in the MRI acquisition. Patients with PD onset at <63 years old were included in the early or middle-onset PD (E-MOPD) group. Patients with PD onset at ≥63 years old were included in the late-onset PD (LOPD) group. The standard age of 63 years old was based on the average age of PD onset in Japanese patients [15]. The age of onset was defined as the time when the patient noticed motor symptoms. The disease duration (years) and levodopa equivalent daily dose (LEDD) [16] were used as indicators of disease progression.

**MRI acquisition.** We performed MRI examinations by using a 1.5T magnetic resonance system (Magnetom ESSENZA Siemens Healthcare, Erlangen, Germany). The acquisition parameters were as follows: 3D magnetization-prepared rapid acquisition gradient echo, head matrix coil, 800 ms inversion time, 1700 ms repetition time, 4 ms time to echo, 12˚ fractional anisotropy, 230 × 230 field of view, 256 × 256 matrix size, and 1.2 mm slice thickness. MRI examinations were performed 119 times (minimum: once; maximum: five times per patient).

**MRI processing.** We used MacOS Mojave version 10.14.6 for MRI image processing. FreeSurfer (version 6.0; Massachusetts General Hospital, Harvard Medical School) was used to estimate and analyze the cortical thickness from 3D T1 images. First, we created an automated cortical model as a preprocessing step. Cortical model creation processing includes motion correction, averaging of multiple T1 images, removal of non-brain tissue, automated Talairach transformation, segmentation of subcortical and deep gray matter volume structures, signal

value normalization, gray matter boundary tessellation, automated geometric correction, maximum signal value turning defined as the transition to other tissues, and surface deformation based on signal value gradients that optimally locate gray and white matter boundaries and gray matter cerebrospinal fluid boundaries. The process for estimating the cortical thickness from a complete cortical model includes the registration of the cortical model to a sphere, cortical segmentation and labeling, and the estimation of the cortical thickness from soft and white matter surfaces. The obtained data were subjected to quality control according to the ENIGMA Cortical Quality Control Protocol 2.0 (http://enigma.ini.usc.edu/protocols/imaging-protocols/). No participants were excluded because of excessive superficial or subcortical segmentation errors.

## Statistical analyses

**Demographic variables.** We performed the demographic and clinical statistical analyses using JMP 13.2.0 (SAS Institute Inc., Cary, NC, USA). We tested for group differences in the demographic and clinical variables between the E-MOPD and LOPD groups by using Mann–Whitney pairwise comparisons or Pearson's chi-squared test. Statistical significance was set at $p < 0.05$.

**Cortical thickness analyses.** Statistical analysis of the cortical thickness was performed using a general linear model in Qdec (http://surfer.nmr.mgh.harvard.edu/fswiki/FsTutorial/QdecGroupAnalysisV6.0) embedded in FreeSurfer. We examined the areas of cerebral cortical thinning in relation to disease progression. Additionally, we investigated between-group differences in the slope of regression lines between the cortical thickness and disease progression. These represented areas had more rapid thinning with disease progression in LOPD than in E-MOPD. S1 Fig provides a schematic of this concept. To remove the effect of age, we used the age of onset as a covariate in the correlation analysis between the clinical course and cortical thickness in each group. The age during MRI imaging was used as a covariate in the intergroup comparison of the correlation between the cortical thickness and clinical course in both groups. Statistical results were corrected for multiple comparisons per cluster by using the Monte Carlo method. All resulting cortical areas were significantly different ($p < 0.05$).

# Results

## Demographic variables and clinical information during MRI imaging

There were no significant differences in the demographic variables or clinical information between the E-MOPD and LOPD groups except for age. Table 1 summarizes the demographic variables and clinical information during MRI.

There were no significant differences in the demographic variables or clinical information between the groups, except that the E-MOPD group was younger than the LOPD group.

## Correlation between cortical thickness and disease progression

**Correlation between cortical thickness and disease progression in all patients.** By using the disease duration as an index of disease progression, we observed significant negative correlations between the cortical thickness and disease progression in the bilateral superior frontal gyrus, middle frontal gyrus, precentral gyrus, postcentral gyrus, paracentral lobule, and right occipital lobe ($p < 0.05$) (Fig 1). Similarly, by using LEDD as an index of disease progression, we recorded significant negative correlations between the cortical thickness and disease progression in the bilateral precentral gyrus, paracentral lobule, superior parietal lobule, precuneus, and occipital lobule ($p < 0.05$) (Fig 1). S1 Table outlines the identified clusters. The

**Table 1. Demographic variables and clinical information during MRI imaging.**

| Variable | E-MOPD (n = 31) | | | | | LOPD (n = 31) | | | | | Test stats, P-value |
|---|---|---|---|---|---|---|---|---|---|---|---|
| Sex, male, n (%) | 15 (48.3) | | | | | 15 (48.3) | | | | | 0.6113 |
| | **Mean** | **SD** | **Median** | **Min** | **Max** | **Mean** | **SD** | **Median** | **Min** | **Max** | |
| Age, y | 62.7 | 6.8 | 64 | 46 | 72 | 74.6 | 4.1 | 74 | 67 | 86 | < .0001 |
| Disease duration, y | 6.6 | 3.9 | 7.1 | 0.5 | 15.3 | 5.5 | 2.9 | 5.2 | 1.1 | 11.4 | 0.375 |
| Age of onset, y | 55.7 | 6.0 | 57 | 42 | 63 | 69.1 | 4.4 | 68 | 64 | 79 | |
| LEDD, mg | 532.4 | 271.8 | 510 | 0 | 1069 | 476.7 | 286.3 | 450 | 0 | 1335 | 0.331 |

LEDD, levodopa equivalent daily dose; SD, Standard deviation; E-MOPD, early or middle-onset Parkinson's disease; and LOPD, late-onset Parkinson's disease.

cluster information includes coordinates of the Montreal Neurological Institute (MNI) displaying the maximum -log10 p-value in the identified cluster, cortical area (anatomical location of MNI coordinates), cluster size (mm$^2$), and clusterwise p-value (p-value for each cluster).

**Correlation between cortical thickness and disease progression in E-MOPD and LOPD.** When using the disease duration as an index of disease progression, there were significant negative correlations between the cortical thickness and disease progression in the superior frontal gyrus, middle frontal gyrus, central precentral gyrus, central posterior gyrus, and bilateral paracentral lobule of the E-MOPD and in the superior frontal gyrus, middle frontal gyrus, central precentral gyrus, central posterior gyrus, paracentral lobule, precuneus, and bilateral occipital lobe bilaterally of the LOPD group (p < 0.05) (Fig 2). When using LEDD as an index of disease progression, there was no significant correlation in the E-MOPD group (Fig 2). By contrast, the LOPD group displayed significant negative correlations between the cortical thickness and disease progression in the superior frontal gyrus, middle frontal gyrus, precentral gyrus, postcentral gyrus, paracentral lobule, superior parietal lobule, precuneus, occipital lobe, and left superior temporal gyrus on both sides (p < 0.05) (Fig 2). S2 and S3 Tables present information on the identified clusters.

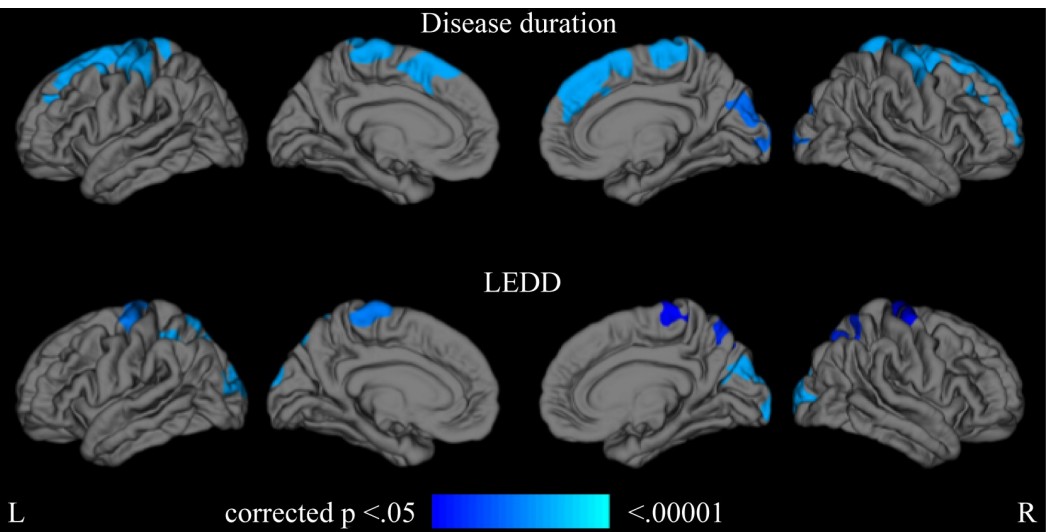

**Fig 1. Areas of cortical thinning with disease progression in all patients.** Color maps indicate areas with significant thinning (p < 0.05). LEDD, levodopa equivalent daily dose.

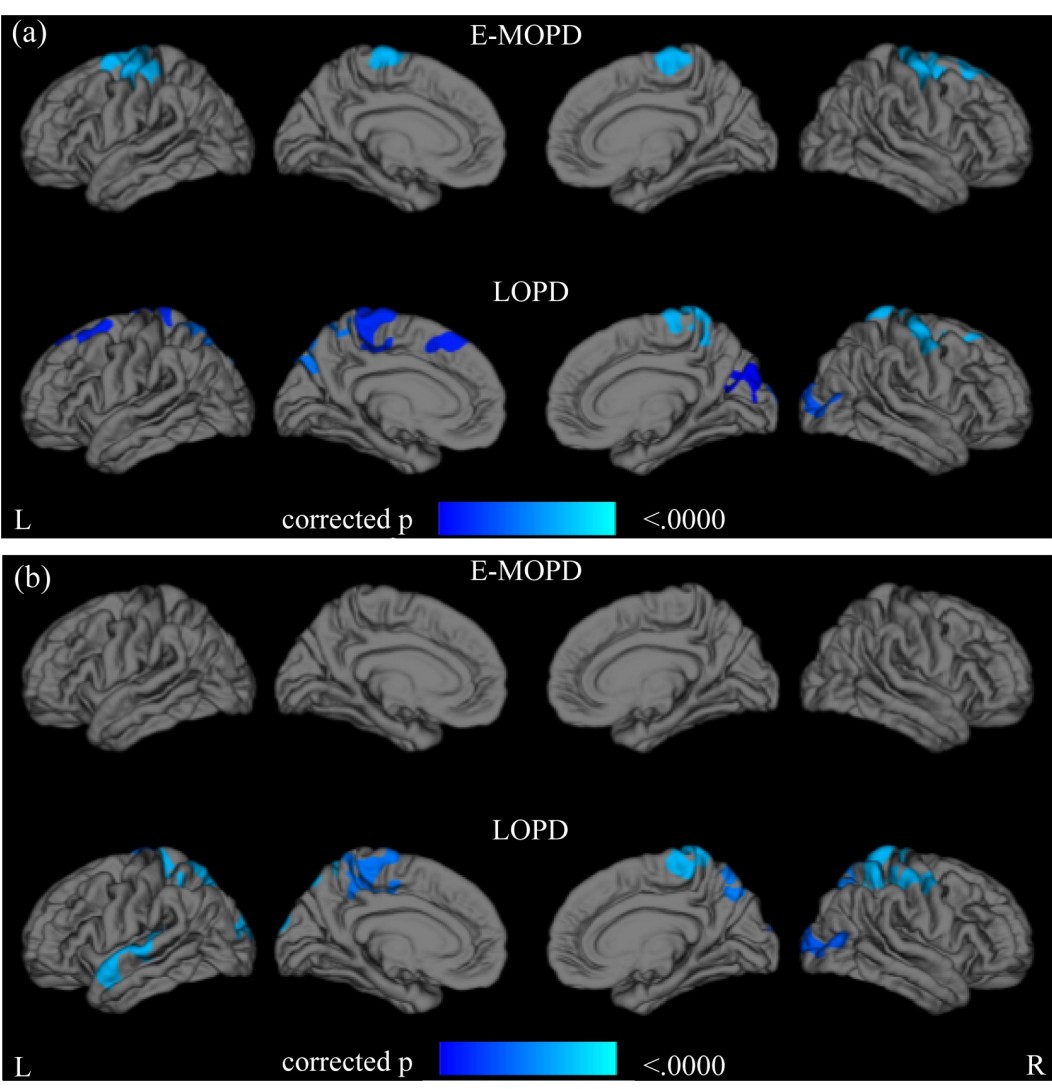

**Fig 2. Correlation between cortical thickness and disease progression in E-MOPD and LOPD.** (a) Areas of cortical thinning with prolonged disease duration. (b) Areas of cortical thickness thinning with an increase in LEDD. LEDD, levodopa equivalent daily dose; E-MOPD, early or middle-onset Parkinson's disease; and LOPD, late-onset Parkinson's disease.

**Between-group differences in the slope of regression lines between cortical thickness and disease progression.** When using the disease duration as an index of disease progression, the precuneus displayed more rapid cortical thinning in the LOPD group than in the E-MOPD group (p < 0.05) (Fig 3). When using LEDD as an index of disease progression, the bilateral superior temporal gyrus, paracentral lobule, superior parietal lobule, occipital lobule, right superior frontal gyrus, middle frontal gyrus, temporal pole, left precentral gyrus, left postcentral gyrus, and precuneus revealed more rapid cortical thinning in the LOPD group than in the E-MOPD group (p < 0.05) (Fig 3). There were no areas of rapid cortical thinning in the E-MOPD group compared with the LOPD group. S4 Table presents information on the identified clusters.

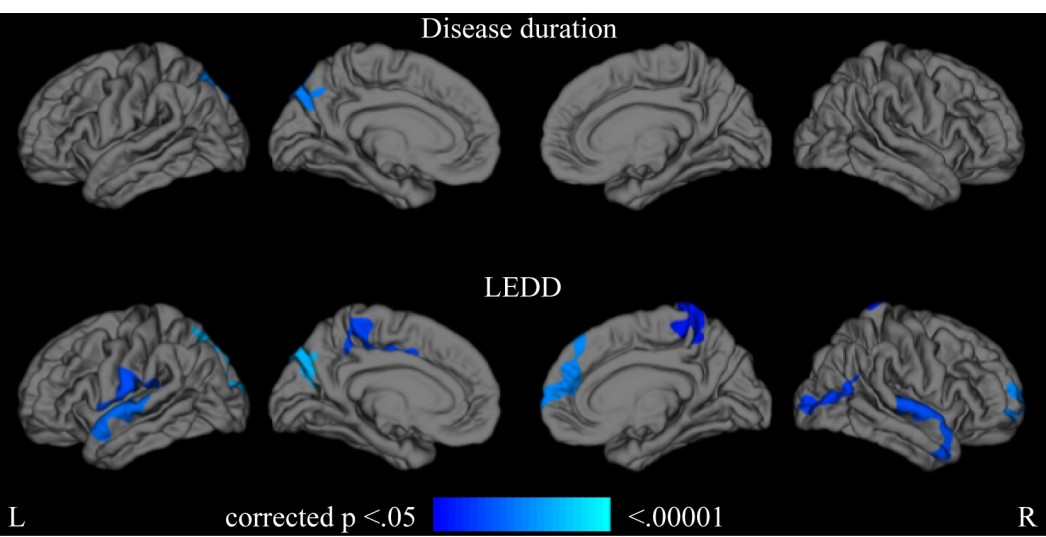

**Fig 3. Areas of rapid cortical thinning in the LOPD and E-MOPD groups (p < 0.05).** LEDD, levodopa equivalent daily dose; E-MOPD, early or middle-onset Parkinson's disease; and LOPD, late-onset Parkinson's disease.

## Discussion

This is the first study to capture age-dependent differences in cortical thinning during disease progression in PD. The LOPD group displayed more extensive cortical thinning with disease progression than the E-MOPD group.

Several reports have been published on the relationship between the clinical symptoms of PD and cortical thinning. In terms of motor symptom severity, bradykinesia is associated with the thinning of the parietotemporal sensory association area [17]. Postural instability is associated with thinning of the cuneus and precuneus [18], whereas tremor is associated with the thinning of a wide range of cortical regions, including the bilateral middle frontal gyrus, left orbitofrontal cortex, bilateral cingulate gyrus, left middle temporal gyrus, left lateral temporal gyrus, left inferior parietal lobule, and left occipital gyrus [19]. Among non-motor symptoms, cognitive dysfunction and psychiatric symptoms are closely related to cerebral cortex atrophy. PD with mild cognitive impairment (PD-MCI) is characterized by the thinning of the parietal and occipital cortices compared with PD without dementia. When combined with the thinning of the frontal cortex, PD-MCI reportedly progresses to PD with dementia [20–23]. Patients with PD-MCI without the thinning of the parietal and occipital lobes recover normal cognitive function, but those with thinning of the aforementioned areas do not [21]. Cognitive domain–specific reports have demonstrated that attention and executive function deficits were correlated with the thinning of frontal regions [22,24]. Moreover, visuospatial cognitive deficits were correlated with the thinning of the parietal and occipital lobes [20]. Radziunas et al. [25] and Kubera et al. [26] mentioned that impulse control disorder is correlated with the thinning of the inferior frontal gyrus, superior frontal gyrus, anterior cingulate gyrus, and precentral gyrus in PD. A study of the correlation between psychological symptoms and cortical thinning in patients with PD, as assessed by the Caregiver-Administered Neuropsychiatric Inventory, demonstrated that nocturnal behavior and irritability were correlated with frontal cortex thinning, aggression and agitation with temporal lobe thinning, and apathy with insular cortex thinning [27].

In this study, we found that the LOPD group had cortical regions with more rapid thinning than the E-MOPD group. Among all cortical areas that displayed more rapid cortical thinning

in the LOPD group, the superior temporal gyrus, temporal pole, paracentral lobule, superior parietal lobule, precuneus, and occipital lobe were reportedly associated with bradykinesia [17], the precuneus and occipital lobe were associated with postural instability [18,19], and extensive cortices from the frontal to the parietal and occipital lobes were associated with cognitive dysfunction [20–23]. The pattern of cortical thinning observed in the current study may be explained by the pathophysiology underlying the clinical features of elderly patients with PD, such as α-synuclein deposition in elderly patients [12]. The serum level of uric acid, which is an antioxidant, is reportedly low in patients with PD; this situation may be attributed to the weakening of the neuroprotective mechanisms of the body [28]. In addition to α-synuclein deposition, amyloid-β deposition is also observed in older patients with PD and may contribute to the difference in thinning depending on the age of onset [29,30]. The cause of rapid PD progression in elderly patients has not yet been clarified. However, this clarification will likely lead to the development of therapies that inhibit disease progression.

A limitation of this study was its retrospective nature because it made the acquisition of motor and non-motor symptoms difficult in a standardized manner. Therefore, we were unable to analyze the relationship between cortical thinning and the characteristics of motor symptoms, such as tremor, bradykinesia, and postural instability, as well as non-motor symptoms, such as cognitive dysfunction and autonomic symptoms. Although it is desirable to standardize the number of MRI scans for patients, this study had a retrospective design and could not be standardized.

## Conclusion

Patients with older-onset PD demonstrated more extensive areas of reduced cortical thickness compared with those with early and middle-onset PD. Furthermore, patients with older-onset PD had cortical areas with more rapid thinning. Therefore, brain structural MRI with automated cortical thickness analysis may provide clues for a better understanding of age-related structural changes in patients with PD in an objective manner, which may be related to the underlying pathophysiology.

## Supporting information

**S1 Fig. A schematic image depicting the difference in the slope of regression lines between the E-MOPD and LOPD groups.** E-MOPD, early or middle-onset Parkinson's disease; LOPD, late-onset Parkinson's disease.
(DOCX)

**S1 Table. Information on the identified clusters in Fig 1.**
(DOCX)

**S2 Table. Information on the identified clusters in Fig 2(A).**
(DOCX)

**S3 Table. Information on the identified clusters in Fig 2(B).**
(DOCX)

**S4 Table. Information on the identified clusters in Fig 3.**
(DOCX)

## Acknowledgments

We would like to thank Editage (www.editage.com) for the English language editing.

## Author Contributions

**Conceptualization:** Kazuhide Seo, Ichiro Matunari, Toshimasa Yamamoto.

**Data curation:** Kazuhide Seo.

**Formal analysis:** Kazuhide Seo.

**Funding acquisition:** Kazuhide Seo.

**Investigation:** Kazuhide Seo.

**Methodology:** Kazuhide Seo, Ichiro Matunari.

**Project administration:** Kazuhide Seo.

**Resources:** Kazuhide Seo, Ichiro Matunari, Toshimasa Yamamoto.

**Software:** Kazuhide Seo.

**Supervision:** Kazuhide Seo, Ichiro Matunari, Toshimasa Yamamoto.

**Validation:** Kazuhide Seo.

**Visualization:** Kazuhide Seo.

**Writing – original draft:** Kazuhide Seo.

**Writing – review & editing:** Kazuhide Seo, Ichiro Matunari, Toshimasa Yamamoto.

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
