## [Decision Letter · Decision Letter 0]

11 Aug 2022

PONE-D-21-33205Cerebral cortical thinning in Parkinson's disease depends on the age of onsetPLOS ONE

Dear Dr. seo,

Thank you for submitting your manuscript to PLOS ONE. After careful consideration, we feel that it has merit but does not fully meet PLOS ONE’s publication criteria as it currently stands. Therefore, we invite you to submit a revised version of the manuscript that addresses the points raised during the review process. Although the Reviewers had conflicting decisions on your manuscript, I have opted to render a decision of Major Revisions. Please carefully address the points that have been raised. In addition to those listed by the Reviewers, there were a few aspects that were not clear to me. For example, was the enrollment period really nearly 20 years long? If so, it seems likely that there would have been major scanner changes throughout the course of the study. It is not clear how this would have affected the results. In addition, it is not clear what was done for patients with multiple scans, since the work presented seems to be cross-sectional in nature.

We look forward to receiving your revised manuscript.

Kind regards,

Niels Bergsland

Academic Editor

PLOS ONE

Journal Requirements:

Reviewers' comments:

Reviewer's Responses to Questions

**Comments to the Author**

1. Is the manuscript technically sound, and do the data support the conclusions?

Reviewer #1: Partly

Reviewer #2: Partly

2. Has the statistical analysis been performed appropriately and rigorously? 

Reviewer #1: I Don't Know

Reviewer #2: No

3. Have the authors made all data underlying the findings in their manuscript fully available?

Reviewer #1: Yes

Reviewer #2: No

4. Is the manuscript presented in an intelligible fashion and written in standard English?

Reviewer #1: No

Reviewer #2: Yes

5. Review Comments to the Author

Reviewer #1: The aim of the present study is to identify cortical regions with different thinning depending on the age of onset in patients with PD. Sixty-two patients with PD were included in this study. The present manuscript is interesting. However, I have some concerns:

1. This appears to be a retrospective study, have the authors any intention of conducting a prospective one more aligned with their hypotheses?. How did the authors arrive at the study size of 62? It was not clear from the text.

2. It is not clear why the authors divided the PD group in younger and older. Why authors have opted to 63 years and not 40 years, 55 years?

3. An important concern is about the fact that the authors did not compare PD groups to age- and sex-matched healthy controls.

4. The specificity of these imaging results in terms of clinical meaning or physiopathology of PD is unclear: age or other factors may explain the results.

5. There is little mention of the existing extensive literature about the changes and progression of expression of the morphological and functional changes in the brain related to the age of onset in PD with both MRI and PET.

Reviewer #2: Appendix tables arent properly labeled.

The correlation presented with disease duration has no context as far as duration from diagnosed or personal notice.

Were there any other clinical information, such as UPDRS or Hoehn-Yahr correlated with the significant differences? Otherwise the impact is limited.

6. PLOS authors have the option to publish the peer review history of their article (what does this mean?). If published, this will include your full peer review and any attached files.

Reviewer #1: No

Reviewer #2: No

---

## [Author Response · Author response to Decision Letter 0]

22 Dec 2022

I wish to re-submit the manuscript titled “Cerebral cortical thinning in Parkinson’s disease depends on the age of onset.” The manuscript ID is PONE-D-21-33205.

We thank you and the reviewers for your thoughtful suggestions and insights. The manuscript has benefited from these insightful suggestions. I look forward to working with you and the reviewers to move this manuscript closer to publication in the PLOS ONE.

The manuscript has been rechecked, and the necessary changes have been made in accordance with the reviewers’ suggestions. The responses to all comments have been prepared and attached herewith/given below. The changes made to the manuscript are indicated in red font color. 

Thank you for your consideration. I look forward to hearing from you.

Sincerely,

---

## [Decision Letter · Decision Letter 1]

7 Feb 2023

Cerebral cortical thinning in Parkinson’s disease depends on the age of onset

PONE-D-21-33205R1

Dear Dr. seo,

We’re pleased to inform you that your manuscript has been judged scientifically suitable for publication and will be formally accepted for publication once it meets all outstanding technical requirements.

Kind regards,

Niels Bergsland

Academic Editor

PLOS ONE

Additional Editor Comments (optional):

Reviewers' comments:

Reviewer's Responses to Questions

**Comments to the Author**

1. If the authors have adequately addressed your comments raised in a previous round of review and you feel that this manuscript is now acceptable for publication, you may indicate that here to bypass the “Comments to the Author” section, enter your conflict of interest statement in the “Confidential to Editor” section, and submit your "Accept" recommendation.

Reviewer #1: All comments have been addressed

2. Is the manuscript technically sound, and do the data support the conclusions?

Reviewer #1: Partly

3. Has the statistical analysis been performed appropriately and rigorously? 

Reviewer #1: I Don't Know

4. Have the authors made all data underlying the findings in their manuscript fully available?

Reviewer #1: Yes

5. Is the manuscript presented in an intelligible fashion and written in standard English?

Reviewer #1: No

6. Review Comments to the Author

Reviewer #1: (No Response)

7. PLOS authors have the option to publish the peer review history of their article (what does this mean?). If published, this will include your full peer review and any attached files.

Reviewer #1: No

---

## [Editor Report · Acceptance letter]

10 Feb 2023

PONE-D-21-33205R1 

Cerebral cortical thinning in Parkinson’s disease depends on the age of onset 

Dear Dr. seo:

I'm pleased to inform you that your manuscript has been deemed suitable for publication in PLOS ONE. Congratulations! Your manuscript is now with our production department. 

Kind regards, 

on behalf of

Dr. Niels Bergsland 

Academic Editor

PLOS ONE